# The ER membrane complex (EMC) can functionally replace the Oxa1 insertase in mitochondria

**Büsra Güngör**[1], **Tamara Flohr**[1], **Sriram G. Garg**[2], **Johannes M. Herrmann**[1]*

**1** Cell Biology, University of Kaiserslautern, Kaiserslautern, Germany, **2** Institute for Molecular Evolution, Heinrich-Heine-Universität Düsseldorf, Düsseldorf, Germany

* hannes.herrmann@biologie.uni-kl.de

## Abstract

Two multisubunit protein complexes for membrane protein insertion were recently identified in the endoplasmic reticulum (ER): the guided entry of tail anchor proteins (GET) complex and ER membrane complex (EMC). The structures of both of their hydrophobic core sub-units, which are required for the insertion reaction, revealed an overall similarity to the YidC/Oxa1/Alb3 family members found in bacteria, mitochondria, and chloroplasts. This suggests that these membrane insertion machineries all share a common ancestry. To test whether these ER proteins can functionally replace Oxa1 in yeast mitochondria, we generated strains that express mitochondria-targeted Get2–Get1 and Emc6–Emc3 fusion proteins in Oxa1 deletion mutants. Interestingly, the Emc6–Emc3 fusion was able to complement an *Δoxa1* mutant and restored its respiratory competence. The Emc6–Emc3 fusion promoted the insertion of the mitochondrially encoded protein Cox2, as well as of nuclear encoded inner membrane proteins, although was not able to facilitate the assembly of the Atp9 ring. Our observations indicate that protein insertion into the ER is functionally conserved to the insertion mechanism in bacteria and mitochondria and adheres to similar topological principles.

## Introduction

Membranes of bacteria and eukaryotic cells contain different protein translocases. These pore-like structures transport unfolded polypeptides across membranes and, in case of membrane proteins, laterally integrate them into the lipid bilayer [1]. Examples are the SecY/Sec61 complexes of the bacterial inner membrane and the endoplasmic reticulum (ER) [2,3], the beta barrel-structured outer membrane translocases of bacteria, mitochondria and chloroplasts [4,5], and the translocases of the mitochondrial inner membrane (TIM23 and TIM22 complexes) [6,7]. These translocases belong to distinct nonrelated protein families and developed independently during evolution.

Protein translocation can also be mediated by a second group of translocation machineries that do not form defined pores but rather facilitate protein translocation by local distortion

**Data Availability Statement:** All relevant data are within the paper and its Supporting information files.

**Funding:** This project was funded by grants from the Deutsche Forschungsgemeinschaft (DIP

MitoBalance and HE2803/9-1) and the
Landesschwerpunkt BioComp (all to JMH). The
funders had no role in study design, data collection
and analysis, decision to publish, or preparation of
the manuscript.

**Competing interests:** The authors have declared
that no competing interests exist.

**Abbreviations:** Cox2, subunit 2 of cytochrome
oxidase; EMC, ER membrane complex; ER,
endoplasmic reticulum; ERAD, ER-associated
degradation; GET, guided entry of tail anchor
proteins; MTS, matrix targeting sequence.

and compression of lipid bilayers [8]. Such a mechanism was recently proposed for the ER-associated degradation (ERAD) pathway machinery [9].

Locally distorted and compressed lipid bilayers are also used by insertases, which integrate hydrophobic proteins into membranes. Substrates of these insertases include membrane proteins that lack large hydrophilic regions on the trans-side of the membrane (such as in the case of tail-anchored proteins) or multispanning membrane proteins whose more complex topogenesis relies on the cooperation of insertases with a canonical protein translocase.

The mitochondrial protein Oxa1 was discovered in the early 90s as the first representative of these insertases [10,11] and served as the founding member of the YidC/Oxa1/Alb3 family. These closely related and functionally exchangeable proteins [12–15] mediate membrane insertion of proteins into the inner membranes of bacteria and mitochondria as well as in the thylakoid membrane of chloroplasts [16–23]. Several YidC structures were published recently, which suggest that these monomeric proteins accelerate the partitioning of hydrophobic segments into the lipid bilayer [24–27].

Two recently identified protein complexes serve as insertases in the ER membrane: the guided entry of tail anchor proteins (Get)1–Get2 (in vertebrates WRB-CAML) complex, which facilitates the insertion of tail-anchored proteins [28], and the ER membrane complex (EMC), a multimeric insertase consisting of 8 (yeast) or 9 (animals) subunits that acts independently of as well as in conjunction with the Sec61 translocon in the topogenesis of multispanning ER proteins [29–33]. The structures of both complexes were recently solved [34–36], revealing a striking similarity of the reaction centers formed by Get2–Get1 and Emc6–Emc3 with the architecture of YidC/Oxa1/Alb3 proteins despite very limited sequence similarity. On the basis of their structural architecture, it was proposed that all these insertases are members of one related group of proteins, which was named the Oxa1 superfamily [37].

In this study, we report that the mitochondrial Oxa1 protein can be functionally replaced by the core components of the EMC complex, at least in respect to its role in the membrane insertion of proteins. Unlike Oxa1, EMC is, however, unable to facilitate the assembly of the $F_o$ sector of the ATPase, presumably because it is not recognized by mitochondrion-specific assembly factors [38,39]. Our study shows that the EMC complex of the ER and the Oxa1 insertase of mitochondria fulfill analogous molecular functions, consistent with their proposed structural similarity.

## Results

### The core components of the various membrane insertases display similar topology despite limited sequence identity

The YidC, Alb3, and Oxa1 insertases of bacteria, chloroplasts, and mitochondria are characterized by 5 conserved transmembrane domains (Fig 1A). The DUF106 protein family of archaea was proposed to be a distant relative, although it only has 3 transmembrane domains that show similarity to the transmembrane domains 1, 2, and 5 of Oxa1. It was suggested that the DUF106 family gave rise to Emc3 and Get1 on the basis of very similar overall structural organization [34,35,40].

To assess a potential phylogenetic relationship among these proteins, we screened for potential related proteins of Oxa1, Alb3, YidC, DUF106, Emc3, and Get1 and identified 460 unique homologs across eukaryotes and prokaryotes (see S1–S3 Data). Phylogenetic trees were calculated based on trimmed alignments (Fig 1B). These trees supported the common origin of Oxa1, Alb3, and YidC very well and also indicated good bootstrap support for their relationship with members of the DUF106, Get1, and Emc3 families (S1 Fig, S1–S3 Data). Even though the similarity of individual proteins is low, the inclusion of the large number of sequences

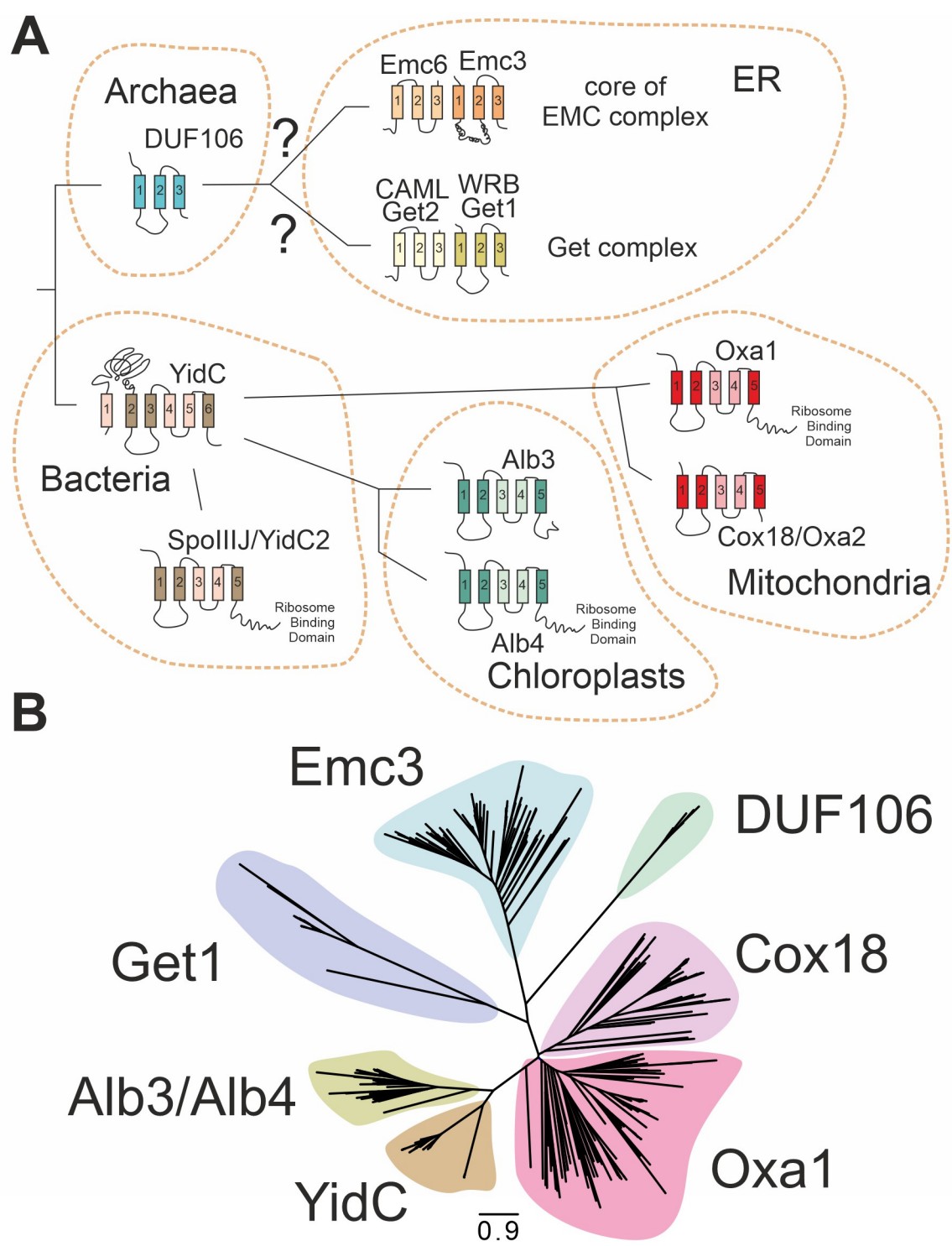

**Fig 1. The members of the Oxa1 superfamily are structurally similar; however, their phylogenetic relationship is unclear. (A)** All shown proteins serve as membrane insertases. Whereas the representatives of bacteria, mitochondria, and chloroplasts are monomeric proteins with 5 or 6 transmembrane domains, the insertases of the ER form oligomeric complexes of 2 (Get1/Get2) or multiple (EMC) subunits. Recent studies on the structure of the GET and EMC complexes proposed that Emc3 and Get1 are structurally related to YidC, Oxa1, and Alb3. The transmembrane domains that share structural similarity are indicated in darker color. **(B)** Blast searches identified 460 homologs of DUF106, Emc3, Get1, Alb3/Alb4, Oxa1, and YidC proteins from different species (see S1 Fig and S1–S3 Data). This tree supports the relatedness of these protein groups and is consistent with the hypothesis that Emc3 is related to a

DUF106-like ancestor protein. While YidC/Oxa1/Alb3 proteins form a closely related group, the branches to Get1, Emc3, and DUF106 are much longer and, thus, their sequences are likely more deviated. ER, endoplasmic reticulum; EMC, ER membrane complex; GET, guided entry of tail anchor proteins.

allowed the construction of a well-supported tree that supports the relatedness of these different groups of insertases. Even if analogy based on convergent evolution cannot be formally excluded, homology based on common ancestry appears more likely.

## A mitochondria-targeted EMC core restores respiration of *Δoxa1* cells

The recently solved structures of the EMC and GET complexes [34–36] suggested that their core centers, consisting of Emc6–Emc3 and Get2–Get1, respectively, resemble the structural organization of YidC. This inspired us to clone the respective regions of Emc6–Emc3 and Get2–Get1 into fusion proteins that also contained the matrix targeting sequence (MTS) of Oxa1 to ensure mitochondrial import of these proteins, the carboxyl-terminal ribosome binding domain of Oxa1 necessary for its interaction with the mitochondrial translation machinery [41–44], as well as a short Oxa1-derived linker to confer insertion into the inner membrane (Fig 2A, S2A and S2B Fig). However, all transmembrane regions of these fusion proteins were derived from ER proteins.

These fusion proteins, which we named mito-EMC and mito-GET, were expressed in *Δoxa1* and *Δcox18* cells, thus in mutants lacking Oxa1 or its paralog Cox18. Surprisingly, the expression of mito-EMC partially suppressed the growth defect of *Δoxa1* mutants even upon growth on the nonfermentable carbon source glycerol (Fig 2B–2D) but not that of *Δcox18* cells (Fig 2B, S2C Fig). The carboxyl-terminal ribosome binding domain of Oxa1 was crucial for this complementation (Fig 2E), consisting with the observation that deletion of this domain in Oxa1 results in a respiration-incompetent phenotype [41,42]. Moreover, point mutations in essential residues of the Emc3 subunit [36] resulted in noncomplementing mito-EMC variants (Fig 2F). Thus, functionality of the EMC subunits was necessary for their ability to take over the function of the Oxa1 insertase in mitochondria. We did not observe any suppression by expression of the mito-GET protein, suggesting that this protein might not be properly integrated into the inner membrane or generally is unable to fully replace mitochondrial insertases in function.

## Mito-EMC can facilitate the insertion of nuclear-encoded inner membrane proteins

Oxa1 mediates the insertion of nuclear-encoded inner membrane proteins that use the so-called conservative insertion pathway (Fig 3A). In order to test whether mito-EMC can take over this function of Oxa1, we isolated mitochondria from *Δoxa1* cells that expressed mito-EMC or Oxa1 for control. Radiolabeled precursor proteins of different model substrates were incubated with these mitochondria to allow their import and intramitochondrial sorting. Then, mitochondria were reisolated and treated with proteinase K to remove nonimported material or converted to mitoplasts by hypoosmotic rupturing of the outer membrane (swelling) and treated with proteinase K. Radiolabeled proteins that were integrated into the inner membrane became protease accessible, giving rise to characteristic fragments (Fig 3B and 3C, S2D and S2E Fig; white arrowheads). From the ratio of these fragments to the protease-inaccessible species (black arrowheads), the insertion efficiency of these membrane proteins can be estimated (Fig 3D). These in vitro import experiments showed that mito-EMC (and Oxa1) facilitated membrane insertion of the model proteins Oxa1 [45] and Su9(1–112)-DHFR [46]. Previous studies have shown that the Oxa1-mediated insertion efficiency is dependent on the

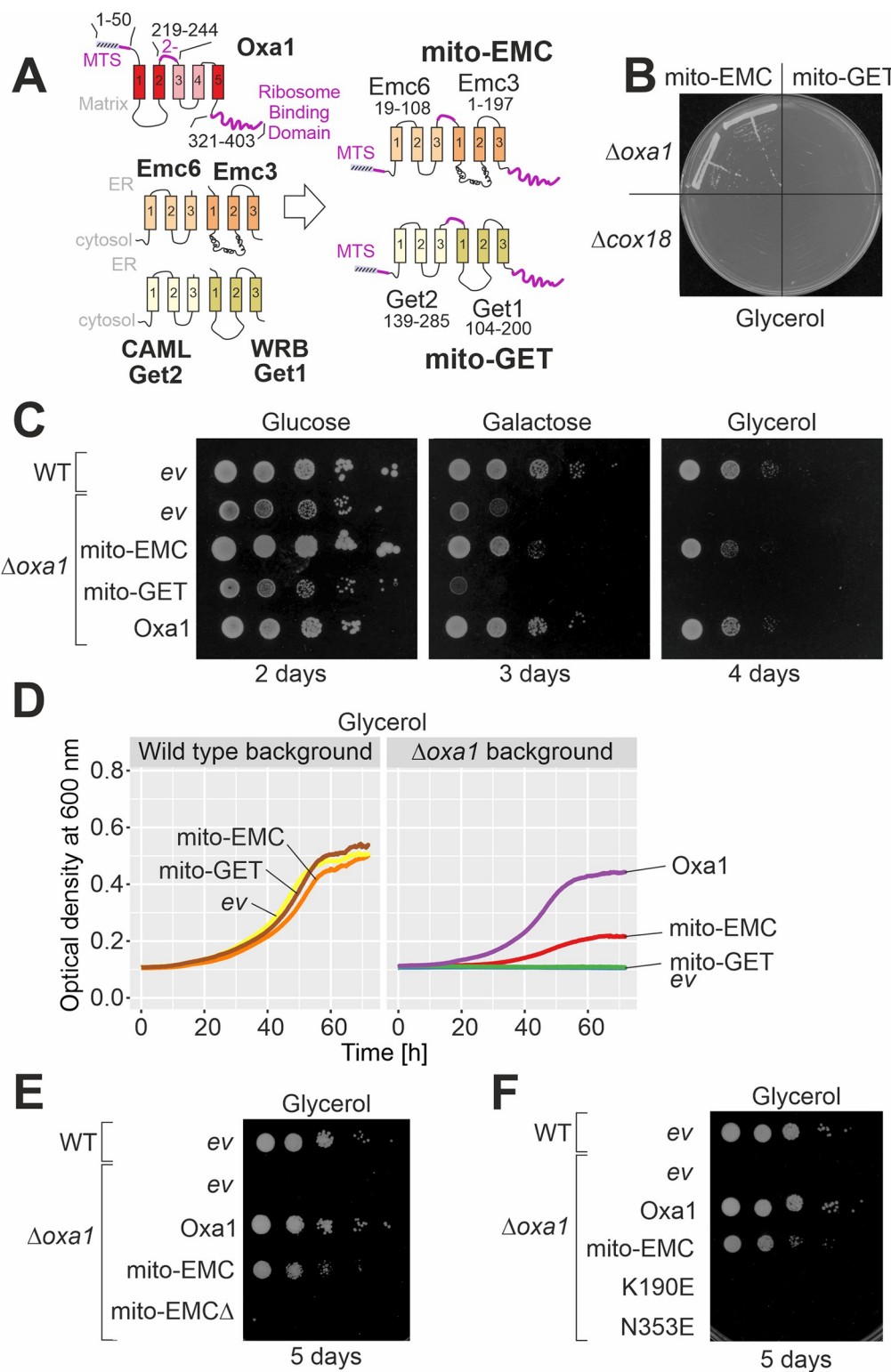

**Fig 2. Expression of a mitochondria-targeted Emc6–Emc3 fusion protein (mito-EMC) suppresses the growth defect of a Δoxa1 mutant.** **(A)** Schematic representation of the fusion proteins used in this study. The segments shown in purple are derived from Oxa1 in order to ensure import and insertion of the proteins into the inner membrane. **(B)** Simple growth test on glycerol plates on which cells can only grow if they generate energy from respiration. **(C)** Cells were grown in galactose medium, from which 10-fold serial dilutions were dropped onto plates with the respective

carbon sources. An empty vector (*ev*) control is shown for comparison. **(D)** Growth curves of *Δoxa1* cells with the indicated expression plasmids. Shown are mean values of 3 technical replicates (*n* = 3). The underlying data can be found in the S4 Data. **(E, F)** Growth test on glycerol plates with mito-EMC mutants that lack the carboxyl-terminal ribosome-binding domain of Oxa1 or that carry mutations in functionally relevant residues of Emc3. EMC, ER membrane complex; GET, guided entry of tail anchor proteins; MTS, matrix targeting sequence; WT, wild type.

negative charge in the region of protein substrates that are translocated to the intermembrane space and that positively charged regions are not exported by Oxa1 [47]. The same charge dependence was also found for mito-EMC–mediated insertion, suggesting that both proteins facilitate membrane insertion by a comparable mechanism that drives the negatively charged sequence to the positively charged side of the inner membrane [27,48].

## Mito-EMC can facilitate the membrane insertion of mitochondrial translation products

Next, we assessed the insertion of mitochondrially encoded proteins. Particularly, the insertion of the subunit 2 of cytochrome oxidase (Cox2) is Oxa1 dependent, and, therefore, Cox2 is not detectable in extracts of *Δoxa1* mutants in western blots (Fig 4A). Expression of the mito-EMC, however, restored the accumulation of the Cox2 protein almost to wild-type levels. To detect the levels of newly synthesized mitochondrial translation products, we monitored mitochondrial protein synthesis in whole cells after inhibition of cytosolic translation by cycloheximide (Fig 4B). This showed that *Δoxa1* cells that contained Oxa1 or mito-EMC were able to synthesize proteins, whereas no translation products were observed in cells expressing mito-GET, suggesting that the mitochondrial DNA was lost in these cells. The increased depletion of mitochondrial DNA was reported before in Oxa1-deficient cells [49].

The synthesis of mitochondrial translation products can be monitored in isolated mitochondria to which radiolabeled $^{35}$S-methionine is added [50]. Cox2 is initially produced as a precursor protein with an N-terminal leader sequence which is cleaved after membrane insertion. The accumulation of the Cox2 precursor is characteristic for *Δoxa1* mitochondria (Fig 4C), owing to their membrane insertion deficiency [51]. Mitochondria containing mito-EMC instead of Oxa1 showed no or only low levels of Cox2 precursors (Fig 4D). However, they showed strongly reduced amounts of Atp9, both in its monomeric form and in its SDS-resistant Atp9 oligomer.

When mitochondria were extracted with salt buffer or carbonate, most translation products were found in the membrane fractions. The mitochondrially encoded ribosomal protein Var1 was, as expected, found in the carbonate supernatant (Fig 4E). In the mito-EMC sample, also some Cox2 precursor was recovered in the carbonate supernatant (white arrowhead). This indicates that mito-EMC, just as Oxa1, efficiently mediates the insertion of mitochondrial translation products into the inner membrane.

## Mito-EMC fails to promote the assembly of the $F_oF_1$ ATPase

Since we observed considerably reduced levels of oligomeric Atp9 in the mito-EMC samples, we carried out immunoprecipitation experiments with antibodies against Oxa1 or ATPase subunits. When mitochondria were lysed after the labeling reaction and Oxa1 and mito-EMC were isolated by immunoprecipitation with antibodies against the carboxyl terminus of Oxa1, subunits of the ATPase were only coisolated with Oxa1 but not with mito-EMC (Fig 4F). When the labeling was stopped and isolated mitochondria were further incubated, monomeric Atp9 became converted into the oligomeric form in the presence of Oxa1, but to a much lesser degree in the mito-EMC mutant (S3A–S3C Fig). Moreover, immunoprecipitation of ATPase

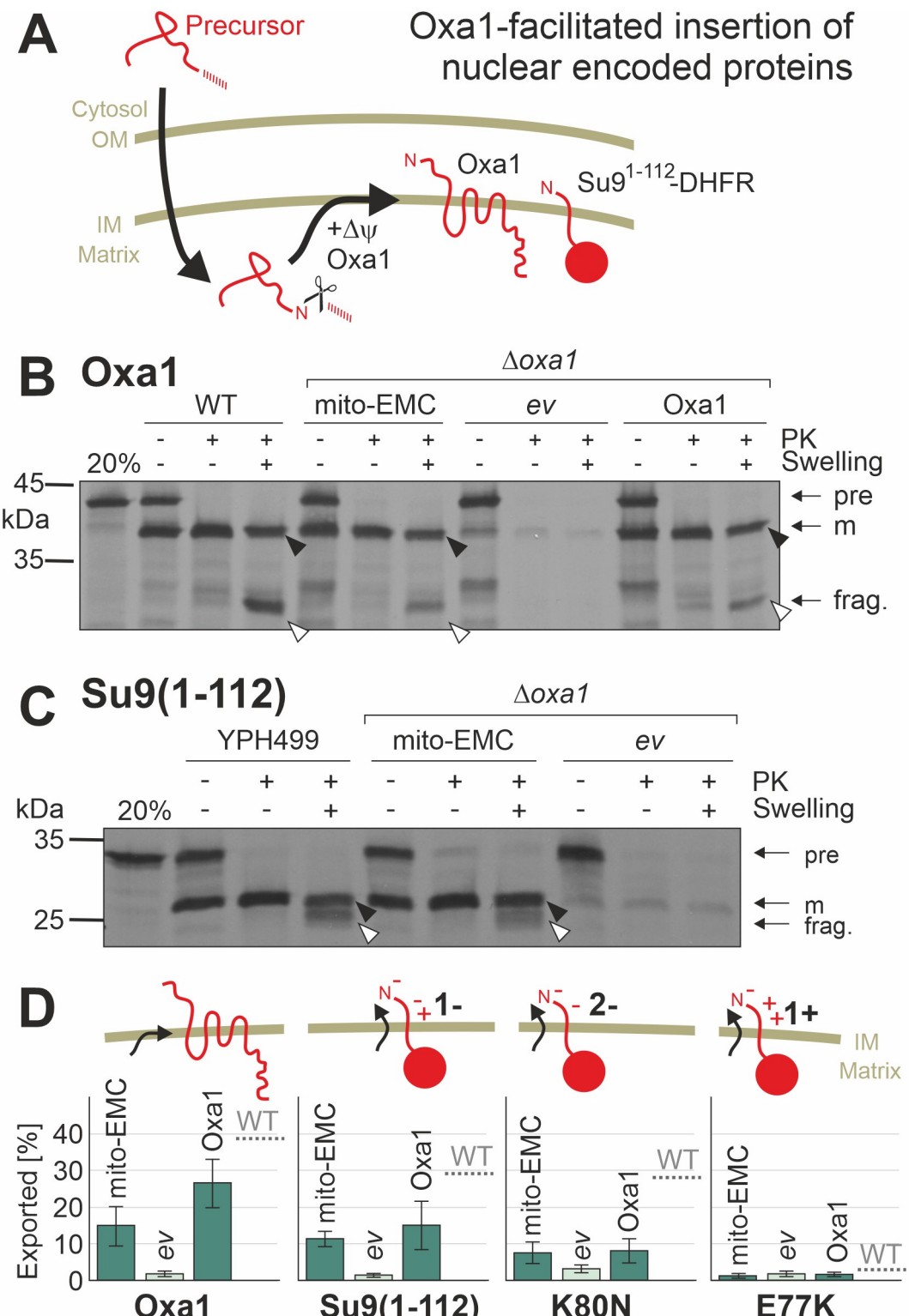

**Fig 3. Protein insertion by mito-EMC, like that by Oxa1, depends on negative charges in the transferred sequence. (A)** Schematic representation of the import and Oxa1-mediated membrane insertion of nuclear-encoded proteins. The scissors indicate the proteolytic removal of the N-terminal mitochondrial targeting sequence by the matrix processing peptidase. Protein insertion depends on Oxa1 and the membrane potential (Δψ). **(B, C)** Radiolabeled Oxa1 and Su9(1–112)-DHFR were incubated with mitochondria isolated from the indicated mutants for 30 minutes at 25°C. Samples were divided into 3 fractions

and treated with or without PK under isosmotic or hypoosmotic (Swelling) conditions. Upon swelling, protease treatment generates a fragment (frag., white arrowhead) from the membrane-embedded protein, but not from the translocation intermediate that still resides in the matrix (black arrowhead). **(D)** Insertion efficiencies of Oxa1, Su9(1–112)-DHFR and two Su9(1–112)-DHFR point mutants with more negative (K80N) or positive charge (E77K) of its intramitochondrial space region were quantified. Mean values and standard deviations of 3 replicates are shown. The underlying data can be found in S4 Data. EMC, ER membrane complex; PK, proteinase K; WT, wild type.

complexes with antibodies against Atp1, Atp2, or the F1-part of the ATPase copurified Atp6, Atp8, and Atp9 in Oxa1-containing mitochondria, whereas in mitochondria of the mito-EMC mutant, only Atp6 and Atp8, but no Atp9, was coisolated (Fig 4G). From this, we conclude that mito-EMC can insert Cox2 into the inner membrane. However, it does not efficiently promote the insertion and/or assembly of the Atp9 oligomer in the inner membrane. This is also confirmed by considerably diminished ATPase activity levels in mito-EMC mitochondria (Fig 4H). Thus, whereas mito-EMC shares the insertase activity with Oxa1, it seems to be limited in its ability to interact with mitochondrial ATPase assembly factors (Fig 4I and 4J).

## Discussion

The EMC complex of the ER was identified only rather recently [29]. The range of its physiological activities is not entirely clear, in part because its ability to insert proteins into the ER membrane overlaps with that of the Sec61 translocase [33]. The essential nature of the Sec61 complex makes it difficult to elucidate the EMC activity in the absence of that of the dominating translocon. The recently solved structure and the identification of a rather small Emc6–Emc3 core region of the EMC complex inspired us to try an in vivo reconstitution approach in the mitochondrial inner membrane. Our observations on this mito-EMC protein allow a number of conclusions:

1. Emc3 and Emc6 together indeed form a minimal insertase unit that is able to promote protein insertion into a lipid bilayer. Of course, in the native EMC complex, other subunits might add further activities and properties that are not reflected in the mito-EMC protein. However, the Emc6–Emc3 core region with their 6 transmembrane domains are sufficient to promote protein insertion. This observation confirms the insertion mechanism that was proposed on the basis of recent cryo-electron microscopic studies that revealed that a membrane-embedded hydrophobic groove formed by Emc3 and Emc6 constitutes a transient binding site that incorporates transmembrane stretches into a disturbed and locally thinned lipid bilayer [35–37,52,53].

2. Complexome profiling [30] and proteomic analysis of EMC-deficient cells [54] revealed that EMC substrates are transmembrane proteins, many of which span the ER membrane multiple times. Studies on reconstituted liposomes showed that while the EMC complex is able to integrate proteins of rather simple topology on its own [32], the assistance of the Sec61 translocon is often necessary for accurate topogenesis of multispanning proteins [31]. We observed that mito-EMC is able to mediate the insertion of the mitochondrial translation products that range from proteins of rather simple topology (Atp8 and Cox2) to multipass proteins with several transmembrane domains (Cox1, Cox3, cytochrome *b*, and Atp6). Thus, our in vivo reconstitution indicates that, in principle, the EMC core is able to mediate the insertion of a broad range of membrane proteins in the absence of a Sec61-mediated insertion activity.

3. Oxa1 and YidC mediate the translocation of negatively charged protein segments, but largely fail to export positively charged regions [27,47,48]. The mito-EMC–mediated

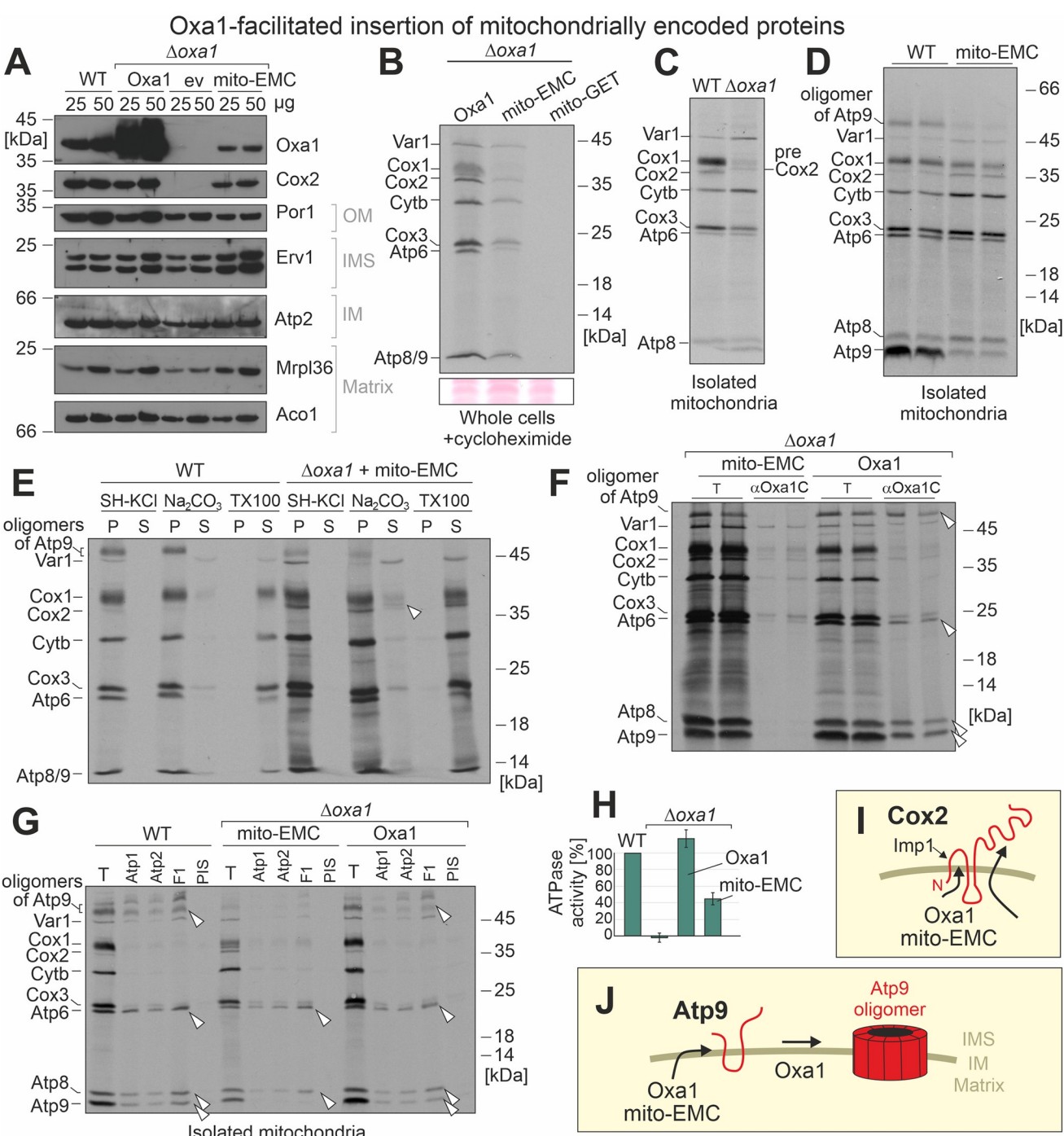

**Fig 4. mito-EMC mediates the insertion of mitochondrial encoded proteins but fails to take over the Oxa1-facilitated assembly of the Atp9 oligomer.**
**(A)** Mitochondria were isolated from the indicated strains. The indicated proteins were detected by western blotting. The Oxa1-specific serum detects Oxa1 as well as the mito-EMC fusion protein. **(B)** Δ*oxa1* mutants expressing Oxa1, mito-GET, or mito-EMC were grown in galactose medium to log phase. Cytosolic translation was inhibited by cycloheximide. $^{35}$S-methionine was added to radiolabel mitochondrial protein synthesis. Cells were harvested and lysed. Proteins were precipitated by trichloroacetic acid and visualized by SDS-PAGE and autoradiography. The pinkish signals on the bottom show Ponceau S-stained protein bands verifying equal loading. **(C, D)** Mitochondria were isolated from the indicated strains and incubated in the presence of $^{35}$S-methionine in translation buffer for 30 minutes at 30°C. Mitochondria were washed and newly synthesized proteins visualized by autoradiography. The precursor form of Cox2 is labeled as preCox2. **(E)** Translation products were radiolabeled in organello before mitochondria were incubated with buffer (SH-KCl), carbonate (Na$_2$CO$_3$), or detergent (TX100), respectively, to extract proteins from the membranes. Samples were centrifuged and split into P and S. Proteins in the S were precipitated with trichloroacetic acid. Protein signals were visualized by autoradiography. **(F, G)** Mitochondrial translation products were radiolabeled for 30 minutes before mitochondria were lysed with Triton X-100. The extract was either directly loaded onto the gel (T,

equivalent to 10% total) or after immunoprecipitation with antibodies raised against the carboxyl terminus of Oxa1 (αOxa1C), subunits of the ATPase, or with PIS. Arrowheads depict immunoprecipitated ATPase subunits. **(H)** The levels of oligomycin-sensitive ATPase were measured in isolated mitochondria and quantified. Shown are mean values of 3 replicates. The underlying data can be found in S4 Data. **(I, J)** While mito-EMC is able to mediate the membrane insertion of Cox2, it does not efficiently promote the assembly of Atp9 into its oligomeric form. Thus, mito-EMC seems only to be able to take over some of the Oxa1-mediated functions in mitochondria. EMC, ER membrane complex; GET, guided entry of tail anchor proteins; P, pellet; PIS, preimmune serum; S, supernatant; WT, wild type.

insertion adhered to the same properties, again supporting the conclusion that both insertases promote the same type of biochemical reaction (Fig 4I).

4. In addition to its role as an insertase for membrane proteins, Oxa1 was proposed to facilitate the assembly of oligomeric complexes [55,56]. Interestingly, we observed that the mito-EMC failed to promote efficient biogenesis of the Atp9 ring (Fig 4J). Assembly of the $F_0$ sector of the ATPase is a complicated process relying on a number of different assembly factors [57–59]. It was proposed that some of these are recruited by Oxa1 [38], consistent with our observations. However, it should be noted that despite the Atp9 defect of the mito-EMC strain, still about half of the normal ATPase activity accumulated in the mutant explaining why this mutant efficiently grows on nonfermentable carbon sources.

While the functional complementation of *Δoxa1* mutants by mito-EMC alone cannot prove a common evolutionary origin of these insertases, this observation strongly supports this assumption. We suggest that our findings demonstrate that the fundamental mechanisms by which proteins are integrated into the inner membrane of mitochondria and the ER are universal, and the insertion process of membrane proteins adheres to the same principles despite the about 3 billion years of evolution since archaeal and bacterial lineages separated. This similarity may now be used to uncover fundamental principles of EMC function specifically and protein insertion more generally.

## Methods

### Yeast strains and plasmids

The yeast strains used in this study are either based YPH499 (MATa *ura3 lys2 ade2 trp1 his3 leu2*) or for the *Δcox18* experiment based on W303 (MATa *ura3 ade2 trp1 his3 leu2*). The *Δoxa1* [60] and *Δcox18* [12] strains used for this study were described before.

To generate the mito-GET and mito-EMC expression plasmids, the coding regions for the following protein stretches were amplified from genomic DNA and ligated into a pYX232 empty vector plasmid downstream of the *TPI* promoter by using the restriction sites *Xma*I and *Sal*I or *Sal*I and *Nco*I using a Gibson assembly protocol [61]: Oxa1(1–50), Get2(139–285), Oxa1(219–244), Get1(104–200), Oxa1(321–403) for mito-GET and Oxa1(1–50), Emc6(19–108), Oxa1(219–244), Emc3(1–197), and Oxa1(321–403) for mito-EMC. The mito-EMCΔ variant lacked the Oxa1(321–403) segment and instead contained a triple hemagglutinin tag to allow its detection in western blots. For the mito-EMC K190E and N353E mutants, individual residues of Emc3 were replaced by glutamate residues that correspond to residues 31 or 180 of human or residues 26 or 187 of the yeast Emc3, respectively. These residues were previously identified as being essential for functionality [36].

For control, a pYX232 plasmid was generated that contained the entire Oxa1 reading frame (1–403). All plasmids were verified by sequencing.

Strains were grown at 30˚C in minimal synthetic medium containing 0.67% yeast nitrogen base and 2% glucose, galactose, or glycerol as carbon source. For plates, 1.5% of agar was added to media.

## Drop dilution assay

To test growth on plates, drop dilution assays were conducted. Yeast cells were grown in synthetic galactose media to mid log phase. After harvesting 2 OD (600 nm) of cells and washing with sterile water, a 1:10 serial dilution was prepared in sterile water. Equal amounts of the dilutions were dropped on agar plates to determine growth differences. Pictures of the plates were taken after 2 to 5 days of incubation.

## Growth curve assay

To test growth in liquid media, growth curves were performed. Yeast cells were grown in synthetic galactose media to mid log phase. After harvesting 2 OD (600 nm) of cells and washing with sterile water, cells of OD 0.1 were resuspended in the experimental media and transferred into a clear 96-well plate. Automated OD measurements were performed at 600 nm in the ELx808 Absorbance Microplate Reader (BioTek, Winooski, Vermont, USA). ODs were measured for 72 hours every 10 minutes at 30°C in technical triplicates.

## Isolation of mitochondria

To isolate crude mitochondria, yeast strains were cultivated in synthetic galactose media to mid log phase and harvested by centrifugation (5 minutes, 3,000 g). After washing the pellets with water and centrifugation (5 minutes, 3,000 g), the weight of the cell pellet was determined. Pellets were resuspended in 2 ml per g wet weight in MP1 (100 mM Tris, 10 mM DTT), incubated for 10 minutes at 30°C, and centrifuged again (5 minutes, 3,000 g). Pellets were washed with 1.2 M sorbitol and centrifuged (5 minutes, 3,000 g) before resuspending in 6.7 ml per g wet weight in MP2 (20 mM KPi buffer pH 7.4, 1.2 M sorbitol, 3 mg per g wet weight zymolyase from Seikagaku Biobusiness, Tokyo, Japan) and incubated at 30°C for 60 minutes. The following steps were conducted on ice. After centrifugation (5 minutes, 2,800 g), pellets were resuspended in 13.4 ml/g wet weight in homogenization buffer (10 mM Tris pH 7.4, 1 mM EDTA, 0.2% BSA, 1 mM PMSF, 0.6 M sorbitol), and a cooled glass potter was used to homogenize the sample with 10 strokes. After homogenization, the extract was centrifuged 3 times (5 minutes, 2,800 g) while always keeping the mitochondria-containing supernatant. To pellet mitochondria, samples were centrifuged for 12 minutes at 17,500 g. Pellets were resuspended in SH buffer (0.6 M sorbitol, 20 mM HEPES pH 7.4). The concentration of the purified mitochondria was adjusted to 10 mg/ml protein. Aliquots were snap frozen in liquid nitrogen and stored at −80°C.

## Import of radiolabelled precursor proteins into mitochondria

To prepare radiolabelled ($^{35}$S-methionine) proteins for import experiments, the TNT Quick Coupled Transcription/Translation Kit from Promega (Walldorf, Germany) was used according to the instructions of the manufacturer. To determine the ability of proteins to be imported into mitochondria, in vitro import assays were conducted. Mitochondria were resuspended in a mixture of import buffer (500 mM sorbitol, 50 mM HEPES pH 7.4, 80 mM KCl, 10 mM Mg(OAc)$_2$, 2 mM KPi) with 2 mM ATP and 2 mM NADH to energize them for 10 minutes at 25°C. The import reaction was started by addition of the radiolabelled lysate (1% final volume). Import was stopped after 30 minutes by transferring the mitochondria into cold SH buffer (0.6 M sorbitol, 20 mM HEPES pH 7.4) or 20 mM HEPES pH 7.4 (swelling conditions). The remaining precursors outside of the mitochondria were removed by protease treatment (PK) for 30 minutes. Moreover, 2 mM PMSF was added to stop protein degradation. After centrifugation (15 minutes, 25,000 g, 4°C), the supernatant was removed. Pellets were resuspended in SH buffer containing 150 mM KCl and 2 mM PMSF (SH-KCl) and centrifuged

again (15 minutes, 25,000 g, 4°C). Pellets were then lysed in sample buffer (2% sodium dodecyl sulfate, 10% glycerol, 50 mM dithiothreitol, 0.02% bromophenolblue, 60 mM Tris/HCl pH 6.8) and heated to 96°C for 5 minutes. Samples were run on a 16% SDS-gel, blotted onto a nitrocellulose membrane, and visualized with autoradiography.

## Labeling of mitochondrial translation products (in organello and in vivo)

Translation products were labeled in isolated mitochondria as described previously [62]. Mitochondria (100 μg protein) were incubated in translation buffer (0.6 M sorbitol, 150 mM KCl, 15 mM $KH_2PO_4$, 13 mM $MgSO_4$, 0.15 mg/ml of all amino acids except methionine, 4 mM ATP, 0.5 mM GTP, 5 mM α-ketoglutarate, 5 mM phosphoenolpyruvate, 3 mg/ml fatty acid-free bovine serum albumin, 20 mM Tris/HCl pH 7.4) containing 0.6 U/ml pyruvate kinase and 10 μCi $^{35}$S-methionine. Samples were incubated for indicated time points at 30°C, and labeling was stopped by addition of 25 mM unlabeled methionine. The samples were further incubated for 3 minutes to complete synthesis of nascent chains. Mitochondria were isolated by centrifugation, washed in 1 ml 0.6 M sorbitol, 20 mM HEPES/HCl, pH 7.4, lysed in 20 μl sample buffer, and subjected to SDS-PAGE.

In vivo labeling of mitochondrial translation products was performed in whole cells in the presence of cycloheximide essentially as described [63]. Proteins were precipitated in the presence of 10% trichloroacetic acid and precipitates washed with ice-cold acetone.

## Carbonate extraction

Translation products were radiolabeled in organello. Reactions were then split into 3 aliquots. The aliquots were centrifuged (10 minutes, 16,000 rpm, 4°C) and the supernatant discarded. Into one, 500 μl SH-KCl was added, one was treated with 500 μl 0.1 M $Na_2CO_3$ to extract integral membrane proteins, and one was treated with 500 μl 0.1% TX100. The samples were kept on ice for 30 minutes and mixed multiple times. After centrifugation (10 minutes, 18,000 rpm, 4°C), the supernatant was transferred into a new reaction tube. The pellet was centrifuged shortly (5 seconds) to discard remaining liquids and lysed in 25-μl sample buffer for 5 minutes with agitation. Proteins in the supernatant were precipitated with 150 μl 72% trichloroacetic acid for 30 minutes on ice. Samples were then centrifuged (10 minutes, 18,000 rpm, 4°C), the pellet washed with 500 μl ice-cold acetone, and centrifuged again (10 minutes, 18,000 rpm, 4°C). The pellet was dried for 10 minutes at 37°C with an open lid. Then, the pellet was lysed in 25-μl sample buffer for 5 minutes with agitation. Samples were run on a 16% SDS gel, blotted onto a nitrocellulose membrane, and visualized by autoradiography.

## ATPase activity assay

To determine the activity of the ATPase, 10-μl isolated mitochondria (100-μg protein) were dissolved in 90-μl lysis buffer (0.1% TX100, 300 mM NaCl, 5 mM EDTA, 10 mM HEPES/KOH pH 7.4) for 10 minutes and centrifuged (10 minutes, 16,000 rpm, 4°C). The supernatant was split into 3 aliquots. One remained untreated, one was treated with 70 μg/ml oligomycin to inhibit the $F_o/F_1$-ATPase function, and one was boiled at 96°C for 2 minutes to assess the phosphate background. A total of 10 μl protein of each of these aliquots was transferred into 180-μl sample buffer (30 mM HEPES/KOH pH 7.4, 50 mM KCl, 5 mM $MgSO_4$) before the reaction was started by addition of 10 μl 100 mM fresh ATP buffered in 300 mM HEPES/KOH pH 7.4. After 30 minutes at 25°C, 4-μl 70 μg/ml oligomycin and 330 μl of fresh solution 1 (160 mM ascorbic acid, 480 mM HCl, 3% SDS, 0.5% $NH_4MoO_4$) were added, and samples were kept on ice for 10 minutes. Then, 500-μl solution 2 (70 mM sodium citrate, 2% sodium

arsenite, 2% acetic acid) was added, and samples were incubated at 25˚C for 20 minutes. The absorbance was measured at 705 nm.

## Sequence analysis, alignments, and phylogeny

Oxa1, Emc3, Get1, and Cox18 proteins from *Homo sapiens* and *Saccharomyces cerevisiae*, Alb3 and Alb4 from *Arabidopsis thaliana*, YidC from *Escherichia coli*, SpoIIIJ from Bacillus subtilis, and *Methanocaldococcus jannaschii* DUF106 were used as seed sequences (S1 Data) to identify homologs using Diamond BLAST [64] against a database of 150 eukaryotes with complete genomes and RefSeq 2016 database [65]. A total of 460 unique homologs that had a minimum of 25% identity and an e-value of less than 1e-10 were chosen for further analysis (see S2 Data). In case of YidC and SpoIIIJ, the first 50 hits were taken to reduce the number of homologs and compensate for the high number of prokaryotic representatives compared to eukaryotes. Maximum likelihood trees were calculated using IQ-tree [66,67] with standard parameters of a 1000 Ultrafast Bootstraps following an alignment performed using MAFFT [68] and trimming using TrimAl [69] and finally were rooted using MAD [70]. The final alignment is provided in S3 Data.

## Antibodies

Antibodies were raised in rabbits against proteins recombinantly expressed in bacteria and purified by affinity chromatography. The Oxa1-specific antibody was raised against the carboxyl-terminal region of Oxa1 that is present in both Oxa1 and mito-EMC. Antibodies against the mitochondrial ATPase were a gift from Marie-France Giraud from CNRS in Bordeaux.

## Supporting information

**S1 Fig. Complete rooted phylogenetic tree for homologs of DUF106, Emc3, Get1, Alb3/Alb4, Oxa1, and YidC.** A rooted phylogenetic tree composed of sequence homologs of DUF106, Emc3, Get1, Alb3/Alb4, Oxa1, and YidC identified as described in the Methods section. The Boot strap values are shown at the nodes. The tree demonstrates a closer relationship between DUF106, Emc1, and Get1 compared to Cox18, YidC/Oxa1/Alb3. Alignments are provided in S3 Data. EMC, ER membrane complex; GET, guided entry of tail anchor proteins. (TIF)

**S2 Fig. Neither mito-EMC nor mito-GET are able to take over the function of the Oxa1 homolog Cox18. (A)** Mitochondria isolated from the indicated strains were incubated in isosmotic or hypoosmotic (swelling) buffer in the absence or presence of PK. Treatment of PK removes accessible protein segments such as the intermembrane space-exposed N-terminus of Oxa1. Note that the mito-EMC protein is protected against PK-treatment consistent with an $N_{in}$-$C_{in}$ topology of the protein. **(B)** The mito-EMC and the mito-GET fusion proteins are detected in western blotting of whole cell extracts. The mitochondrial protein aconitase (Aco1) served as loading control. **(C)** WT and *Δoxa1* cells were transformed with plasmids for the expression of mito-EMC or mito-GET or with an empty vector (*ev*) for control. Cells were grown on the indicated carbon sources. **(D, E)** Radiolabeled Su9(1–112, K80N)-DHFR and Su9(1–112, E77K)-DHFR were incubated with isolated mitochondria as described in Fig 3C. For quantification, see Fig 3D. EMC, ER membrane complex; GET, guided entry of tail anchor proteins; PK, proteinase K; WT, wild type. (TIF)

**S3 Fig. Assembly of ATPase is not efficiently promoted by mito-EMC. (A)** Scheme of the pulse chase labeling experiment. **(B)** Newly synthesized proteins were radiolabeled in isolated

mitochondria for 30 minutes. Radiolabeling was stopped by addition of an excess of cold methionine, and mitochondria were further incubated (chase) at 30˚C for 0, 10, or 30 minutes. (**C**) Quantification of the experiment shown in B. The underlying data can be found in S4 Data. EMC, ER membrane complex.
(TIF)

**S1 Data. Seed sequences used to search for sequences related to Alb3, Oxa1, YidC, Emc3, and DUF106.** Oxa1, Emc3, Get1, and Cox18 proteins from *Homo sapiens* and *Saccharomyces cerevisiae*, Alb3 and Alb4 from *Arabidopsis thaliana*, YidC from *Escherichia coli*, SpoIIIJ from Bacillus subtilis, and *Methanocaldococcus jannaschii* DUF106 were used as seed sequences. EMC, ER membrane complex.
(MFA)

**S2 Data. Blast hits obtained with the seed sequences.** Listed are the hits that were obtained by the Blast search. A total of 460 unique homologs that had a minimum of 25% identity and an e-value of less than 1e-10 were chosen for further analysis.
(TXT)

**S3 Data. Alignment file for the tree shown in S1 Fig.** The alignment was calculated on basis of maximum likelihood trees as described in Materials and methods.
(ALN)

**S4 Data. Original data on quantified experiments as shown in Figs 2D, 3D and 4H and S3C Fig.** The individual data from quantifications and measurements are shown.
(XLSX)

**S5 Data. Raw images used for this study.** Original scans and images of agar plates are shown here that were used to prepare the figures of this study.
(PDF)

## Acknowledgments

We thank Sabine Knaus, Andrea Trinkaus, and Vanessa Scherer for technical assistance and Maya Schuldiner for discussions and comments on the manuscript. We are very grateful to Marie-France Giraud (CNRS, Bordeaux) for antibodies against ATPase subunits.

## Author Contributions

**Conceptualization:** Büsra Güngör, Tamara Flohr, Sriram G. Garg, Johannes M. Herrmann.

**Data curation:** Büsra Güngör, Tamara Flohr, Sriram G. Garg.

**Formal analysis:** Büsra Güngör, Tamara Flohr.

**Funding acquisition:** Johannes M. Herrmann.

**Investigation:** Büsra Güngör, Tamara Flohr, Johannes M. Herrmann.

**Methodology:** Büsra Güngör, Tamara Flohr, Johannes M. Herrmann.

**Resources:** Johannes M. Herrmann.

**Software:** Büsra Güngör, Tamara Flohr, Sriram G. Garg.

**Validation:** Büsra Güngör, Tamara Flohr, Johannes M. Herrmann.

**Writing – original draft:** Johannes M. Herrmann.

**Writing – review & editing:** Büsra Güngör, Tamara Flohr, Sriram G. Garg.

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
