## [Editor Report · Decision Letter 0]

29 Jul 2021

Dear Dr Herrmann, 

Thank you for submitting your manuscript entitled "The ER transmembrane complex (EMC) can functionally replace the Oxa1 insertase in mitochondria" for consideration as a Short Report by PLOS Biology. Please accept my apologies for the delay in getting back to you as we consulted with an academic editor about your submission.

Your manuscript has now been evaluated by the PLOS Biology editorial staff as well as by an academic editor with relevant expertise and I am writing to let you know that we would like to send your submission out for external peer review.

Please re-submit your manuscript within two working days, i.e. by Jul 31 2021 11:59PM.

Kind regards,

Richard

Richard Hodge, PhD

Associate Editor, PLOS Biology

rhodge@plos.org

PLOS

---

## [Decision Letter · Decision Letter 1]

25 Aug 2021

Dear Dr Herrmann,

Thank you for submitting your manuscript "The ER transmembrane complex (EMC) can functionally replace the Oxa1 insertase in mitochondria" for consideration as a Short Report at PLOS Biology. Your manuscript has been evaluated by the PLOS Biology editors, an Academic Editor with relevant expertise, and by three independent reviewers.

The reviews are attached below. You will see that the reviewers find your manuscript interesting and provocative, but suggest several additional experiments to increase the overall strength of the data to support the conclusions. After discussing these experiments with the Academic Editor, we do think that doing a complementation experiment with the mutated Emc6Emc3 version of the construct, as suggested by Reviewer 2, will be a nice additional control. Moreover, deleting the ribosome-binding segment likely would provide additional information regarding the minimal requirements for complementation. However, while the reciprocal complementation experiment suggested by both Reviewers 2 and 3 seems to be an obvious one, we acknowledge that it might be technically very challenging and, if it doesn't work, it would not invalidate the conclusions of the manuscript. Thus, we won't make this experiment a requirement for publication. 

In light of the reviews, we will not be able to accept the current version of the manuscript, but we would welcome re-submission of a much-revised version that takes into account the reviewers' comments. We cannot make any decision about publication until we have seen the revised manuscript and your response to the reviewers' comments. Your revised manuscript is also likely to be sent for further evaluation by the reviewers.

We expect to receive your revised manuscript within 3 months. 

**IMPORTANT - SUBMITTING YOUR REVISION**

*Re-submission Checklist*

*Published Peer Review*

*PLOS Data Policy*

*Blot and Gel Data Policy*

Sincerely,

Richard

Richard Hodge, PhD

Associate Editor, PLOS Biology

rhodge@plos.org

PLOS

REVIEWS:

Reviewer #1: Major Comments The YidC/Oxa1/Alb3 family of proteins is crucial for membrane insertion in bacteria, mitochondria, and chloroplasts, respectively. Intriguingly, possible homologs of these proteins have recently been discovered in the ER as determined by solving the structures of the ER transmembrane complex (EMC) and the ER guided entry of tail anchor proteins (Get) complex. The structures showed the two Get1-Get2 subunits and the core subunits Emc6-Emc3 of the EMC shared overall similarity to the solved structure of the bacterial YidC.

In this paper submitted by Herrmann and coworkers, the authors test whether the Emc6-Emc3 protein and Get2-Get1 could take over the function of Oxa1 or Oxa2 in mitochondria. Specifically, the constructs had an amino-terminal mitochondrial targeting sequence, a linker region fusing the two subunits (either Get2-Get1 or Emc6-Emc3) and the Oxa1 derived C-terminal (matrix) ribosomal binding domain. Interestingly, the authors showed that the Emc6-Emc3 (but not Get2-Get1) could functionally replace Oxa1 and complement growth in an Oxa1 deletion yeast strain. Neither fusion construct restored growth to the Oxa2 deletion strain. The data showed the mitochondrial inner membrane localized Emc6-Emc3 construct could insert the Oxa1-dependent Oxa1 protein and Cox2. This suggested that Su9 (1-112) was also inserted although the raw data was not entirely convincing (see below). While the Emc6-Emc3 construct did insert some of the known Oxa1 substrates, they conclude that it also fails to function in catalyzing (at least efficiently) the assembly of the Atp9 oligomer. 

Overall, the paper provides clear data that the core subunits Emc6-Emc3 can replace some of the functions of the Oxa1 insertase in mitochondria. Thus it supports the authors' main conclusion that ER protein insertion is functionally conserved to the insertion mechanism in bacteria and mitochondria. 

Other Comments

1.In Figure 3C, the data seems to show little export of the N-tail of Su9 (1-112) with the mito-EMC facilitating insertion while the quantitation of the data presented in 3D suggests that export is quite good. I would recommend showing the replicates of the data used to calculate the bar graphs in 3D in the supplementary data section.

2. In Figure 4A and 4B, there is a band at the position of the Cox2, which the authors indicate is the Cox2 protein, which I assume is the mature form of Cox2. It is hard to judge if there is any precursor Cox2 in the mito-EMC lanes since the predicted position is not indicated. Maybe a darker gel image in the supplementary section would be helpful to rule out there is any precursor. Could the authors indicate where the Cox2 precursor would be located?

3. The authors state on page 8 "immunoprecipitation of ATPase complexes with antibodies against Atp1, Atp2 or the F1-part of the ATPase copurified ATP6, ATP8, and ATP9 in Oxa1-containing mitochondria, but not in those of the mito-EMC mutant." I recommend being more clear in this statement because in the mito-EMC mutant samples, ATP6 and ATP8 are found but not ATP9. Only ATP9 is missing. The authors also confirmed there is diminished ATPase activity (of the F1Fo ATPase) in mito-EMC mitochondria. However, there is still 50% of the ATPase activity when compared to Oxa1 mitochondria. This seems contradictory with the finding that the Atp9 oligomer is not formed in mito-EMC mitochondria. If no ATP9 oligomer forms, should there be no ATPase activity or is the ATPase activity expected to only be reduced? Please comment on this.

4. On page 3 and 4, it is stated that the "YidC monomeric proteins serve as an enzyme that accelerate the spontaneous (though often inefficient, slow and non-productive) partitioning of hydrophobic segments into the lipid bilayer." This is awkward. Obviously, YidC catalyzes insertion but then it is stated there is spontaneous partitioning. Please clarify. Also, YidC membrane insertion can be very efficient; I would omit the phrase "though often inefficient, slow and non-productive".

5. Page 7 (second sentence from top). After the phrase "Oxa1-mediated insertion efficiency" add "of protein substrates". Also, it should be clear that the charge is added to the intramitochondrial space region of Su9 (1-112). 

6. In Figure 4A, lower panel, state what the pinkish bands are? In the 4(C,D) Figure legend, add "onto the gel" after the word "loaded". Should be "loaded onto the gel." 

Reviewer #2: Membrane protein biogenesis relies on different cellular machineries including pore-forming translocons as well as insertases without an evident pore. The mitochondrial inner membrane (IM) contains Oxa1 as an insertase for IM protein insertion from the matrix side. In this manuscript, Güngör et al. report a surprising observation that Emc6 and Emc2, components of the ER insertase EMC, can take the role of Oxa1 when they are expressed as a fused protein together with the Oxa1 C-terminal segment for mitochondrial ribosome recognition and targeted to the mitochondrial IM. The in vitro import and in vitro mitochondrial translation experiments suggested that this artificial fusion protein, mito-EMC, facilitated the insertion of nuclear-encoded Oxa1 and Su9(1-112), and mitochondrially encoded Cox2 into the IM.　The successful substitution of the Oxa1 insertase function with mito-EMC should require integration of mito-EMC in the IM with correct membrane topology, acquiring the correct functional structure of mito-EMC in the IM, correct recognition of Oxa1 substrates by mito-EMC, correct binding of mito-EMC with mitochondrial ribosomes to receive its substrates etc. Thus, to make this provocative finding more convincing, the following points should be clarified.

1. Successful membrane topology formation of mito-EMC is not easily expected, considering the difference in lipid compositions between the ER membrane and IM. Therefore, at least membrane topology, i.e. membrane integration and orientation, of mito-EMC, as expected in the IM should be assessed. 

2. Insertion of the Oxa1 substrates into the IM by mito-EMC should be tested by carbonate extraction, too, in addition to the protease clipping assay.

3. To confirm that the observed insertion of the Oxa1 substrates relies on the mito-EMC function, the effect of the functional mutation in the EMC, such as Emc3(K26L) should be tested.

4. The authors claimed that Emc3 and Emc6 form a minimal insertase unit that is able to promote protein insertion into a lipid bilayer. Then does expression of Emc3 and Emc6 rescue the deletion of the EMC in the ER membrane? In addition, can Oxa1 and Su9 mutant precursor proteins that lack their presequences be inserted into the EM membrane when the EMC deletion is rescued by Emc6-Emc2 in the ER membrane?

5. The requirement of the Oxa1 C-terminal segment for mitochondrial ribosomes should be experimentally tested by comparing the function of mito-EMC with and without the Oxa1 C-terminal segment.

6. If mito-EMC is functional as an insertase in the IM, can EMC substrates artificially containing a mitochondrial presequence be inserted into the IM from the matrix side after import into the mitochondrial matrix?

7. Since the result on mitochondrially targeted Get1-Get2 fusion protein was negative and its reason was not investigated further, this part on mito-GET had better be deleted.

Reviewer #3 (Chris Meisinger, signs his review): The authors identify in this study an interesting aspect of the evolution of membrane protein insertion systems. Based on the recently identified structural similarities of the core insertases of bacteria, ER and mitochondria (which show only little sequence similarities) they reveal here that these insertases have also functional similarities. Central of this study is a in vivo reconstitution approach in the yeast model where they replace the mitochondrial inner membrane insertase Oxa1 with the core subunits of the ER membrane complex (EMC). Mito targeted EMC can (at least partially) rescue respiratory growth in oxa1 deletion strains indicating that an ER insertase can take over the function of a mitochondrial insertase. Moreover, they show then with elegant import assays that mito-EMC can indeed mediate import of typical Oxa1 substrates in organello. However, the assembly function of Oxa1 for the Atp9 ring can not be complemented, indicating that the actual import process is a more conserved function than the role in assembly. 

This is the first study showing that the members of these insertase family can function as protein insertases also outside of their native organelles/environment. This is a very exciting discovery and opens many new aspects for the field for follow up studies.

The study is presented in a very clear and consise manner; the experiments are of high quality and allow to draw the conclusions presented here.

The manuscript is very well suited for a broad audience including the fields of molecular evolution and general protein biogenesis.

I have a few questions that came up during reading the manuscript that might be clarified prior publication.

- Fig. 1: can the archeal member of this family (DUF106) also complement lack of mitochondrial oxa1?

- Could Oxa1 now (the other way round) also complement the EMC complex insertase function? This would be possibly more difficult to test due to the overlap with the Sec61 complex and the lack of a reasonable read out. But maybe there is a way?

- Fig. 4B the authors claim here „Mitochondria containing mito EMC instead of Oxa1 showed no or only low levels of Cox2 precursors". I can not really see this in the Figure. Cox2 seems rather more in the mito-EMC samples. Please clarify.

- how sick is the oxa1 delta strain compared to WT? Would in Figure 2D the growth of WT be faster than of the oxa1 delta strain re-expressing OXA1? This would be interesting to judge the in vivo complementation efficiency (e.g. can the growth rates of left and right graphs be directly compared?). For the in organello experiments (Fig. 3) direct WT comparison was included and looks convincing.

- For the in organello import experiments (Fig. 3) only one timepoint was used and the topology was then tested (which looks convincing). But did the authors also tested and compared the kinetics (different time points) of the protein insertion process?

---

## [Decision Letter · Decision Letter 2]

10 Dec 2021

Dear Dr Herrmann,

Thank you for submitting your revised Short Report entitled "The ER transmembrane complex (EMC) can functionally replace the Oxa1 insertase in mitochondria" for publication in PLOS Biology. I have now obtained advice from the original reviewers and have discussed their comments with the Academic Editor. 

As you can see, the reviewers appreciated the substantial amount of additional data included in the revised manuscript to address their comments. Based on the reviews, we will probably accept this manuscript for publication. Please make sure to address the following data and other policy-related requests that I have provided below (points A-E):

(A) Your manuscript is being considered as a Short Report, which has a maximum of 4 main figure panels. We ask that you please reduce the number of the main figures, either by combining/rearranging the main figures or by moving one of the figures to the supplementary.

(B) In the Financial Disclosure statement in the online submission form, please ensure that you provide all of the grant numbers associated with the grants received from the Deutsche Forschungsgemeinschaft and Landesschwerpunkt BioComp.

(C) You may be aware of the PLOS Data Policy, which requires that all data be made available without restriction: http://journals.plos.org/plosbiology/s/data-availability. For more information, please also see this editorial: http://dx.doi.org/10.1371/journal.pbio.1001797

- Supplementary files (e.g., excel). Please ensure that all data files are uploaded as 'Supporting Information' and are invariably referred to (in the manuscript, figure legends, and the Description field when uploading your files) using the following format verbatim: S1 Data, S2 Data, etc. Multiple panels of a single or even several figures can be included as multiple sheets in one excel file that is saved using exactly the following convention: S1_Data.xlsx (using an underscore).

- Deposition in a publicly available repository. Please also provide the accession code or a reviewer link so that we may view your data before publication. 

Figure 2D, 3D, 5C, S3C 

(D) Please also ensure that each of the relevant figure legends in your manuscript include information on *WHERE THE UNDERLYING DATA CAN BE FOUND*, and ensure your supplemental data file/s has a legend.

(E) We require the original, uncropped and minimally adjusted images supporting all blot and gel results reported in the following figures. 

Figure 3B-C, 4A-E, 5A-B, S2A-B, S2D-E, S3B

We will require these files before a manuscript can be accepted so please prepare and upload them now. Please carefully read our guidelines for how to prepare and upload this data: https://journals.plos.org/plosbiology/s/figures#loc-blot-and-gel-reporting-requirements

------------------

We expect to receive your revised manuscript within two weeks. 

*Published Peer Review History*

*Early Version*

Sincerely,

Richard

Richard Hodge, PhD

Associate Editor, PLOS Biology

rhodge@plos.org

Reviewer remarks:

Reviewer #1: Herrmann and coworkers adequately addressed my concerns in the previous submission. The data shown in the revised paper is clear and supports their conclusions.They show the ER transmembrane complex (Emc6-Emc3) supports the insertion of the tested Oxa1 dependent nuclear encoded proteins (Oxa1 and Su9 derivative), and the insertion of the mitochondrial encoded Cox 2. However, the EMC complex does not support the function of chaperone function of Oxa1 in the assembly of the Fo sector of the F1Fo ATPase. 

Taken together, the report shows that the EMC core region (EMC6-Emc3) can take over some of the functions of the Oxa1 insertase in mitochondria. Thus it supports the authors' main conclusion that ER protein insertion is functionally conserved similar to the insertion mechanism in bacteria and mitochondria. Clearly, this is an important contribution.

Reviewer #2: This is a revised version of the previously submitted manuscript. The authors addressed most of my concerns except for the ones that would take a longer time than the expected time frame. The authors added substantially new data that made the manuscript much stronger and more convincing. This is a nice paper that is sufficiently novel and advances our understanding of the mechanisms of membrane protein targeting and integration into membranes.

Reviewer #3 (Chris Meisinger, signs his review): The authors addressed all issues raised on the previous submission. This is a very exciting study of excellent quality. I support publication in its present form.

---

## [Editor Report · Decision Letter 3]

17 Dec 2021

Dear Dr Herrmann,

On behalf of my colleagues and the Academic Editor, André Schneider, I am pleased to say that we can in principle accept your Short Report "The ER transmembrane complex (EMC) can functionally replace the Oxa1 insertase in mitochondria" for publication in PLOS Biology, provided you address any remaining formatting and reporting issues. These will be detailed in an email that will follow this letter and that you will usually receive within 2-3 business days, during which time no action is required from you. Please note that we will not be able to formally accept your manuscript and schedule it for publication until you have any requested changes.

PRESS

Sincerely,

Richard 

Richard Hodge, PhD 

Associate Editor, PLOS Biology

rhodge@plos.org

PLOS
